EMBO
Molecular Medicine

# Rational design of West Nile virus vaccine through large replacement of 3′ UTR with internal poly(A)

Ya-Nan Zhang[1,†], Na Li[1,†], Qiu-Yan Zhang[1,†], Jing Liu[1], Shun-Li Zhan[2], Lei Gao[2], Xiang-Yue Zeng[1], Fang Yu[1], Hong-Qing Zhang[1], Xiao-Dan Li[1] , Cheng-Lin Deng[1], Pei-Yong Shi[3], Zhi-Ming Yuan[1], Shao-Peng Yuan[2], Han-Qing Ye[1,*]  & Bo Zhang[1,**]

## Abstract

The genus Flavivirus comprises numerous emerging and re-emerging arboviruses causing human illness. Vaccines are the best approach to prevent flavivirus diseases. But pathogen diversities are always one of the major hindrances for timely development of new vaccines when confronting unpredicted flavivirus outbreaks. We used West Nile virus (WNV) as a model to develop a new live-attenuated vaccine (LAV), WNV-poly(A), by replacing 5′ portion (corresponding to SL and DB domains in WNV) of 3′-UTR with internal poly(A) tract. WNV-poly(A) not only propagated efficiently in Vero cells, but also was highly attenuated in mouse model. A single-dose vaccination elicited robust and long-lasting immune responses, conferring full protection against WNV challenge. Such "poly(A)" vaccine strategy may be promising for wide application in the development of flavivirus LAVs because of its general target regions in flaviviruses.

**Keywords** flavivirus; internal poly(A); live-attenuated vaccine; UTR; West Nile Virus
**Subject Categories** Immunology; Microbiology, Virology & Host Pathogen Interaction

## Introduction

The genus *Flavivirus* within the family Flaviviridae comprises a large number of arthropod-transmitted viruses responsible for severe hemorrhagic syndromes or neurological illnesses in humans such as dengue virus (DENV), Zika virus (ZIKV), West Nile virus (WNV), Japanese encephalitis virus (JEV), and yellow fever virus (YFV), posing a serious threat to global health (Koraka *et al*, 2010; Oehler *et al*, 2014; Beckham *et al*, 2016; Cao-Lormeau *et al*, 2016;

Schuler-Faccini *et al*, 2016; Kraemer *et al*, 2017; Van Hoeven *et al*, 2018; Giovanetti *et al*, 2019; Hadfield *et al*, 2019; Fischer *et al*, 2020). Despite the availability of licensed vaccines for human use such as DENV (chimeric live-attenuated), JEV (both inactivated and live-attenuated), TBEV (inactivated vaccine), and YFV (live-attenuated) (Khou & Pardigon, 2017; Fischer *et al*, 2020), efforts are still needed to explore a new general strategy for the development of vaccines against emerging and re-emerging flaviviruses (Khou & Pardigon, 2017; Pierson & Diamond, 2020).

Flaviviruses are enveloped viruses containing an approximately 11 kb, positive-sense, single-stranded RNA genome with an m7GpppAmpN1 cap I structure at the 5′- end but without a poly(A) tail at 3′- terminus (Fig 1A and Appendix Fig S1). The viral genome encodes a single open reading frame (ORF) flanked by highly structured 5′- and 3′- untranslated regions (UTRs) (Appendix Figs S1 and S2). The 5′-UTR (∼ 100 nt) consists of (i) a conserved stem-loop structure (SL-A) that is proposed to be the promoter for viral polymerase recognition and activity and (ii) the cyclization sequence located upstream and downstream of the translation start AUG codon that mediates long-range RNA–RNA interactions (5′UAR, 5′CS, and 5′DAR) (Villordo & Gamarnik, 2009; Roby *et al*, 2014) (Appendix Fig S2). The 3′-UTR, that is longer and more variable (∼ 400–700 nt) than the 5′-UTR, comprises multiple functional RNA elements (Thurner *et al*, 2004; Gritsun & Gould, 2007) involved in viral replication, virulence, and transmission (Durbin *et al*, 2001; Pijlman *et al*, 2008; Villordo & Gamarnik, 2013; Sakai *et al*, 2014; Manokaran *et al*, 2015; Villordo *et al*, 2015; Villordo *et al*, 2016; Filomatori *et al*, 2017; Shan *et al*, 2017).

Comparative alignment of flavivirus 3′-UTRs reveals that RNA structures residing in the upstream portion of 3′-UTR show great variability in their size and shape depending on virus species in addition to a common 3′-stem-loop (sHP-3′-SL) structure containing the complementary cyclization sequences at the terminus of 3′-UTR. For instance, mosquito-borne flaviviruses (MBFV) contain two characteristic structures known as stem-loop (SL) and dumbbell (DB) in their 3′-UTRs which are duplicated, while tick-borne flaviviruses

1 Key Laboratory of Special Pathogens and Biosafety, Wuhan Institute of Virology, Center for Biosafety Mega-Science, Chinese Academy of Sciences, Wuhan, China
2 Beijing Shunlei Biotechnology Co. Ltd., Beijing, China
3 University of Texas Medical Branch, Galveston, TX, USA
  *Corresponding author. Tel: +86 27 87197822; E-mail: yehq@wh.iov.cn
  **Corresponding author. Tel: +86 27 87197822; E-mail: zhangbo@wh.iov.cn
  †These authors contributed equally to this work.

(TBFV) have no DB structure although it possesses two different SLs; classical insect-specific flaviviruses (ISFV) preserve multiple small stem-loop structures without SL or DB structure in MBFV or TBFV, while no known vector flaviviruses (NKFV) include single SL or DB structure (Villordo et al, 2016; Ochsenreiter et al, 2019) (Appendix Fig S1). In contrast to the essential function of sHP-3′-SL element in viral replication (Romero-Lopez & Berzal-Herranz, 2013; Ochsenreiter et al, 2019), the upstream RNA elements serve as viral replication enhancers (Lo et al, 2003; Manzano et al, 2011; Sztuba-Solinska et al, 2013) and mainly play important roles in the regulation of viral virulence (Pijlman et al, 2008), host immune responses (Schuessler et al, 2012; Chang et al, 2013; Manokaran et al, 2015), and host adaptation (Kieft et al, 2015; Filomatori et al, 2017). They thus have become attractive targets for live-attenuated vaccine (LAV) design (Proutski et al, 1997; Durbin et al, 2001; Sakai et al, 2014).

Some attempts to go further by deleting the whole virulence-related SL and DB domains of MBFV failed to get viable viruses (Yun et al, 2009), as we did initially in the context of full-length WNV infectious cDNA clone (Appendix Fig S3). But, interestingly, the loss of replication efficiency of SL + DB null (Yun et al, 2009) or either domain deletion mutants (Liu et al, 2018) could be restored greatly by duplicating adjacent RNA sequence such as from the 3′-terminal region of the viral open reading frame or 3′-UTR (Yun et al, 2009; Liu et al, 2018). Such compensatory strategy by simply duplicating insertions also gives us a hint that the length of nucleotide sequence rather than the primary sequences might be indispensable for efficient replication of flaviviruses. We speculate that impaired replication of WNV mutant lacking SL + DB sequences could be restored by refilling the gap with some straightforward sequences.

In the present study, we used WNV as a model and replaced the SL + DB regions with a poly(A)-stretch (WNV-poly(A)) and explored the feasibility of this strategy for the development of live-attenuated WNV vaccine. After extensive in vitro and in vivo studies, it was demonstrated that WNV-poly(A) is a promising replication-competent, attenuated vaccine candidate, which deserves further clinical development in future study.

## Results

### Rational design and generation of recombinant WNV vaccine through replacement of the entire SL and DB domains with internal poly(A) sequences

As discussed above, we tried to seek universal sequences to restore the viral replication of WNV mutant lacking both SL and DB domains within 3′-UTR for rational design of LAV. It has been found that there are adenylate-rich (A-r) spacers to separate the conserved RNA structures of 3′-UTR in all dengue virus types ($P < 0.05$; Finol & Ooi, 2019). We expanded the analysis to more MBFVs, TBFVs and ISFVs, and the number of these strains was statistically significant. Since the number of NKFV was < 10, it was not analyzed. The sequences alignment reveals that A-r segments with significantly higher adenosine ribonucleotides composition ($P < 0.05$) exist in 3′-UTRs of WNV, JEV, YFV, ZIKV, TBEV, POWV, and CXFV (Appendix Fig S4 and Appendix Table S1–S7), and the A-r segments composed of 6–28 adenosine nucleotides were widely distributed in

structured and unstructured domains of 3′-UTRs. We hypothesize that remote ancestor flaviviruses contain internal poly(A) tract and these A-r segments may represent the evolutionary remnants of long internal poly(A) tract from previous flaviviruses during viral replication and transmission in different hosts. In agreement with our bioinformatic analysis, poly(A)-binding protein (PABP)-binding site has been reported within 3′-UTR of DENV (Polacek et al, 2009). Taken together, we speculate that the impaired replication efficiency caused by SL + DB deletion may be restored by replacing with internal poly(A) tract.

To test our speculation, various lengths of poly(A) sequences were first engineered between NS5 coding region and 3′-sHP-SL to replace the whole SL and DB domains of 3′UTR in the context of the infectious cDNA clone of WNV (Fig 1A). The strategy for the introduction of different lengths of internal poly(A) sequences was depicted as Appendix Fig S5A and described in Material and Method. The length and the location of poly(A) insertion were confirmed by DNA sequencing (Appendix Fig S5B and C). Following the conventional method, a pool of in vitro transcribed full-length genome RNAs with different lengths of internal poly(A) sequences were transfected into BHK-21 cells (Fig 1B). Through several rounds of independent passage of the P0 virus stock (obtained from the supernatants of transfected cells at 120 hpt), increasing number of immunofluorescence assay-positive cells were observed from P0 to P3 (Fig 1B), indicating that viable recombinant viruses were recovered after passage. Complete genome sequencing revealed that the recovered virus comprised an approximately 130 nt of poly(A) tract between NS5 coding region and 3′-sHP-SL and still retained the deletion of the entire SL + DB domain (Fig 1C). To further confirm that the recovered replication-competent virus is rescued by internal poly(A) insertion, we randomly selected three individual plaques from the recovered viruses. The results of sequencing and RT–PCR analysis (Fig 1C and D) demonstrated that all plaque purified viruses contained a similar length of poly(A) tract of $\sim$ 130 nt as recovered passaged virus. Our results confirmed that the loss of replication of SL + DB deletion mutant (Appendix Fig S3B) was rescued by filling the resultant gap with internal poly(A) sequences. In the following studies, the recovered virus was designated as WNV-poly(A).

Analysis of the growth kinetics and plaque morphology of WNV-poly(A) showed that both pooled and purified WNV-poly(A) replicated efficiently with viral titers as high as $\sim$ $10^7$ PFU/ml at 72 h post-infection (hpi), albeit a little lower than those of WT ($10^8$ PFU/ml; Fig 1E), and produced smaller plaques comparing with WT WNV (Fig 1F). Then, the protein composition of purified virions was analyzed by SDS–PAGE gel stained with Coomassie blue. As shown in Fig 1G, the profiles of both capsid (C) and envelope (E) proteins were identical in both purified viral particles of WNV-poly(A) and WT WNV. Consistently, no significant difference in neutralizing activity of anti-sera against WNV was observed between WNV-poly(A) and WT WNV, indicating that WNV-poly(A) had similar antigenic properties to WT virus (Fig 1H). Additionally, both viruses had comparable infectivity, as demonstrated by Western blotting analysis using anti-envelope polyclonal antibody showing similar levels of envelope protein in equal amounts of infectious particles ($10^6$ PFU; Appendix Fig S6A). Overall, our results presented straightforward evidence that internal poly(A) tract could take over the function of the whole SL and DB elements of 3′-UTR for viral replication efficiency.

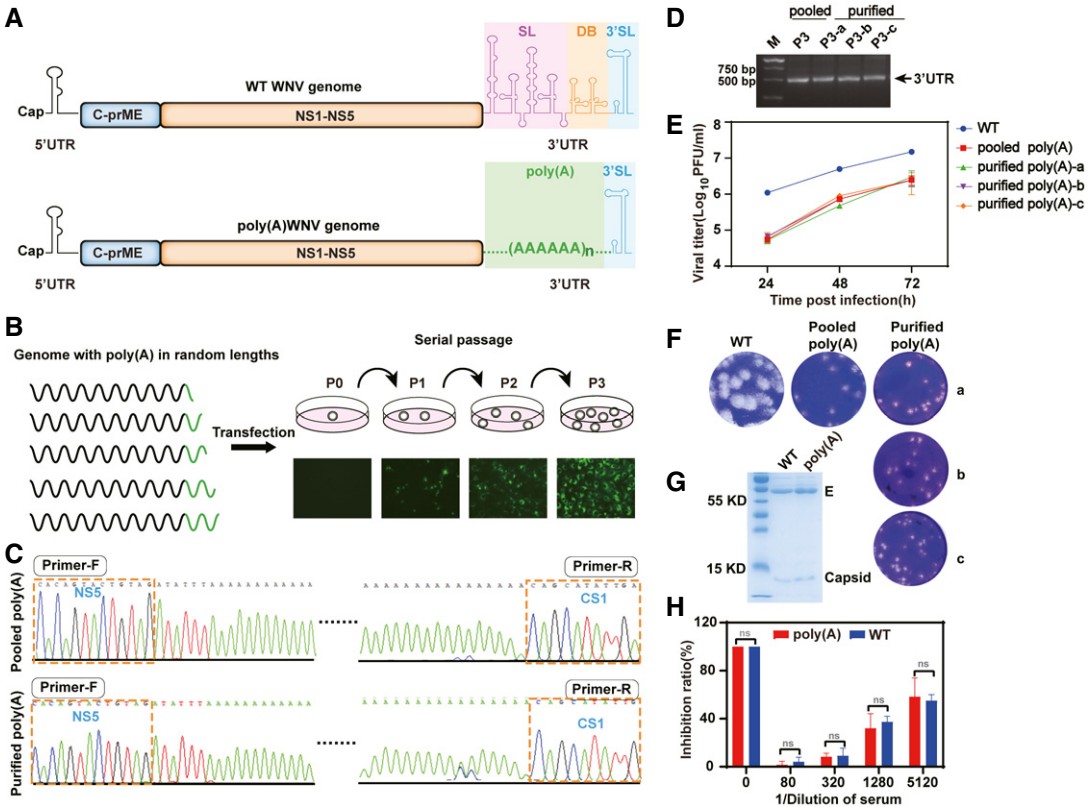

**Figure 1. Development and characterization of WNV-poly(A) vaccine with replacement of internal poly(A) sequences for the entire SL + DB domains.**

A   Schematic representation of WT WNV and WNV-poly(A) genomes. WT 3′UTR is represented by the diagram of secondary structure. Three regions are labeled as SL, stem-loop region; DB, dumbbell region; and 3′SL, stem-loop region, respectively.

B   Rescue of WNV-poly(A) virus. A pool of *in vitro* transcribed recombinant RNAs with different lengths of internal poly(A) tract were electroporated into BHK-21 cells, and the supernatants were blind passaged in BHK-21 cells for three rounds (P1–P3) followed by propagating in Vero cells. Immunofluorescence assay detection of E protein using 4G2 antibody at different passages was shown.

C   Sequence chromatograms of 3′UTR region of P3 WNV-poly(A). Primer-F and Primer-R represent forward and reverse primers, respectively. The primer sequences were described in Materials and methods.

D   Comparison of growth kinetics between WT WNV and WNV-poly(A) viruses in Vero cells.

E   RT–PCR spanning the region from C terminus of NS5 to CS1 region of 3′UTR for P3 pooled WNV-poly(A) and purified viruses. The primer sequences were described in Materials and methods. Two independent experiments in two replicates were conducted and results of a representative experiment were represented. Values represents mean ± SD.

F   Plaque morphology of WT WNV and WNV-poly(A) viruses on BHK-21 cells.

G   SDS–PAGE analysis of purified WT WNV or WNV-poly(A) particles. Bands corresponding to E and Capsid proteins are denoted.

H   Neutralizing activity of anti-sera against WNV was measured between pooled WNV-poly(A) and WT WNV by PRNT assay. Two independent experiments were performed in duplicate. The data shown are the average values of the duplicate from a representative experiment with standard deviation (error bars).

Data information: Statistical analysis was performed with two-way ANOVA. n.s. no statistical differences.
Source data are available online for this figure.

## WNV-poly(A) is highly attenuated in C57BL/6 mice

We next tested the virulence of WNV-poly(A) in mice. 4- to 6-week-old C57BL/6 mice were infected intraperitoneally (i.p.) with $10^4$ PFU (more than 100 times of $LD_{50}$) of WT WNV and different doses of WNV-poly(A) ($10^4$, $10^5$, $10^6$, and $10^7$ PFU), respectively. All the WT WNV-infected mice succumbed within 8 days post-infection (Fig 2A) with apparent weight loss (Fig 2B) and viremia (Fig 2C). In contrast, the mice infected with WNV-poly(A) all survived (Fig 2A) without any visible signs of disease, and the body weights gradually increased during the 14-day observation period (Fig 2B). The 50% lethal dose ($LD_{50}$) of WNV-poly(A) is above $10^7$ PFU, which is

~ 100,000-time higher than that of the WT WNV strain ($LD_{50}$ = 16 PFU) (Zhang *et al*, 2020). It is notable that the viremia was only observed in the mice infected with the highest dose of WNV-poly(A) ($10^7$ PFU) on day one post-infection (Fig 2C). Consistent with the observed disease phenotypes, no significant brain lesions were observed in WNV-poly(A)-infected mice in comparison with mock-infected mice (Fig 2D), while the brain samples from WT WNV-infected mice showed severe vacuolization (black arrows) and inflammation in brain small vessels (yellow arrow) as well as degenerative changes in neurons with cytoplasmic rarefaction (blue arrow) and necrosis with nuclear compaction (red arrow). To further confirm the attenuation of WNV-poly(A), the viral tissue tropism and

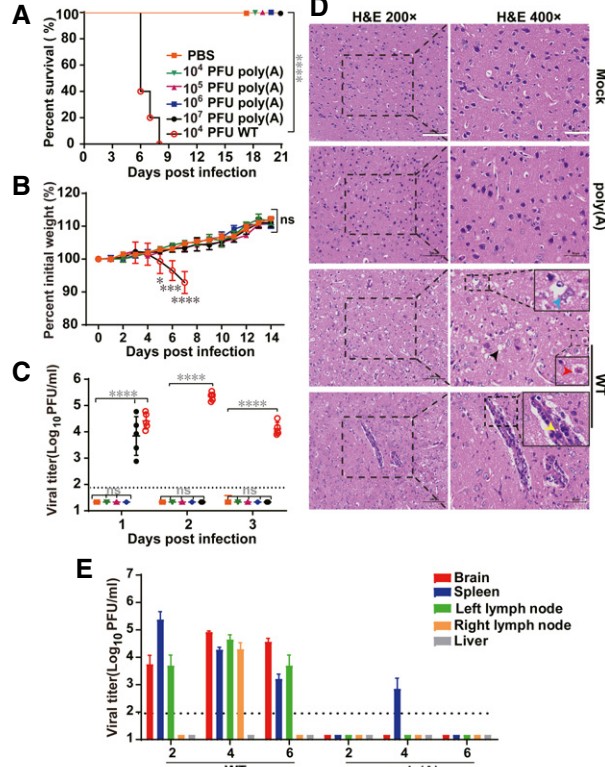

**Figure 2. WNV-poly(A) is highly attenuated in C57BL/6 mice.**

A–D 4- to 6-week-old C57BL/6 mice were injected i.p. with $10^4$–$10^7$ PFU of WNV-poly(A), $10^4$ PFU of WT WNV or PBS (negative control) in a volume of 200 μl. Survival (n = 5) (A) and body weight changes in infected mice were monitored daily (n = 5, the values represent mean ± SD in each group) (B), and viremia was quantified by plaque assay from days 1 to 3 post-infection (n = 5, the values represent mean ± SD in each group, each symbol represents a mouse) (C), Histological analysis of mouse brain. Scale bars represent 100 and 50 μm for left and right panels, respectively. Severe vacuolization (black arrows) and inflammation (yellow arrow) in brain small vessels as well as degenerative changes in neurons with cytoplasmic rarefaction (blue arrow) and necrosis with nuclear compaction (red arrow) were indicated. Scale bars represent 100 and 50 μm for left and right panels, respectively (D).

E Viral load and replication kinetics in different tissues of C57BL/6 mice. Six-week-old female C57BL/6 mice were inoculated subcutaneously with $10^6$ PFU of WT WNV or WNV-poly(A) in the left rear footpad, and tissues were harvested from two mice at the indicated times post-inoculation. Viral loads in brain, spleen, left lymph node, right lymph node, and liver were measured by plaque assay and expressed as PFU per gram of tissue. n = 3, values are mean ± SD, the dotted line corresponds to the limit of detection (LOD).

Data information: Log-rank test was used for (A), two-way ANOVA and one-way ANOVA analysis were used for (B) and (C), respectively. *P < 0.05, ***P < 0.001, ****P < 0.0001, ns represents not significant.

neuro-invasion were examined in C57BL/6 mice following subcutaneous (s.c.) inoculation in left rear footpad with $10^6$ PFU of WT WNV and WNV-poly(A) (Suthar et al, 2013), respectively. Different tissues including brain, spleen, liver, left lymph nodes, and right lymph nodes were collected to measure viral load on days 2, 4, and 6 post-infection (Fig 2E). WT WNV-infected mice had a high viral load in all tested tissues except liver, which is consistent with previous study that liver is resistant to WNV infection (Brown et al, 2007). In

contrast, no virus was detected in the tissues of WNV-poly(A)-infected mice except for the spleen sample from one mouse at 4 days post-infection. These results indicate that WNV-poly(A) is highly attenuated in mouse model without neuro-invasion. To further examine the viral neurovirulence of WNV-poly(A), we infected 4-week-old mice intracranially with different dosages of WNV-poly(A) (Appendix Fig S6B). Despite that the average survival time (AST) of WNV-poly(A) group was 1 day longer than that of WT WNV group, all mice succumbed within 7 days when infected with more than 10 PFU of WNV-poly(A). It is reasonable that WNV-poly(A) retained neurovirulence for mice because it still shares the same envelope antigen, the major neurovirulence determinant (McMinn, 1997), as WT WNV. Thus, much efforts will be devoted to develop improved WNV-poly(A)-based vaccine without neurovirulence by manipulation of the envelope gene in our future work.

**WNV-poly(A) protects C57BL/6 mice from WT WNV challenge**

To assess the potential of WNV-poly(A) as a LAV, 4-week-old C57BL/6 mice were inoculated once with different dosages of WNV-poly(A) ($10^4$, $10^5$, $10^6$, or $10^7$ PFU; Fig 3A). Total IgG antibody and neutralizing antibody titers against WNV were measured with the sera from immunized mice on days 14 and 28 using ELISA assay and PRNT assay, respectively (Fig 3B and C). On day 14, the average IgG titers were 1/800, 1/800, 1/1,650, and 1/3,200 from the $10^4$, $10^5$, $10^6$, and $10^7$ PFU groups, respectively; on day 28, the antibody titers rose to 1/6,500, 1/7,000, 1/8,480, and 1/20,480, respectively. Consistently, the neutralizing antibody titers reached 1/28, 1/40, 1/52, and 1/64 on day 14; and 1/64, 1/120, 1/120, and 1/352 on day 28 from the lowest to the highest doses, respectively. At 30 days post-immunization, the mice from different groups were challenged with a high dose of $3 \times 10^7$ PFU of WT WNV through i.p. route (Fig 3A). All mice immunized with different amounts of WNV-poly(A) were fully protected from challenge and survived without weight loss and detectable viremia (Fig 3D–F). In contrast, all the mice mock immunized with culture media succumbed with severe disease, weight loss, and high levels of viremia. To confirm that WNV-poly(A) functions as a LAV, $10^5$ PFU of WNV-poly(A) and UV-inactivated WNV-poly(A) were used to immunize C57/B6 mice, respectively. Total IgG antibody titers against WNV in the sera from immunized mice were measured on days 14 and 28 using ELISA. There was no detectable antibody reactivity in the mice immunized with UV-inactivated WNV-poly(A) in contrast to seroconversion against WNV observed in the mice immunized with WNV-poly(A) (Appendix Fig S6C), indicating that WNV-poly(A) indeed infected mice to induce antibody response. At the same time, it was found that there was no significant difference between the plaque purified and pooled WNV-poly(A) regarding either the degree of attenuation or the induction of protective immunity in mice (Appendix Fig S7). Overall, these results indicated that WNV-poly(A) functions as a LAV to protect mice from lethal challenge of WT WNV with only one dose of immunization.

To assess T-cell responses in immunized mice, IFN-γ ELISPOT assay was performed to detect the number of WNV-specific IFN-γ secreting cells (n = 4/group) on day 7 after immunization with WNV-poly(A) (Fig 3G). The splenocytes were isolated from the spleen of immunized mice and stimulated with the peptide of CD8+ T-cell epitopes corresponding to WNV envelope and NS4B proteins

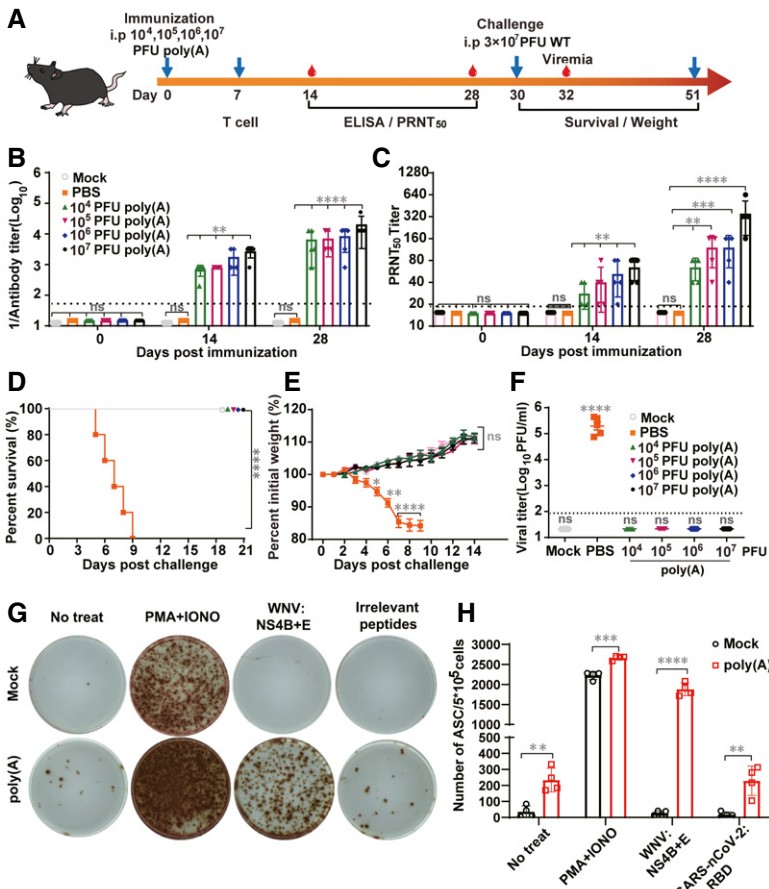

**Figure 3. WNV-poly(A) protects C57BL/6 mice from WT WNV challenge.**

A  Experimental scheme. Groups of 4- to 6-week-old C57BL/6 mice ($n = 5$ per group) were immunized i.p. with $10^4$–$10^7$ PFU of WNV-poly(A), PBS (negative control) or mock (untreated). On day 30 post-immunization, mice were challenged i.p. with $3 \times 10^7$ PFU of WT WNV.

B, C  Total anti-WNV IgG antibody (B) and neutralizing antibody titers (C) in mouse serum at the indicated days post-immunization. (For B and C, the values represent mean $\pm$ SD in each group and the horizontal dotted line represents the limit of detection.)

D–E  Survival (D) and body weight changes were monitored daily after challenge. (The values represent mean $\pm$ SD in each group.) (E).

F  Viremia was quantified by plaque assay on day 2 post-challenge. (The values represent mean $\pm$ SD in each group, each symbol represents a mouse.)

G, H  T-cell responses in vaccinated mice were measured through ELISpot assay. Two groups of C57BL/6 mice ($n = 4$/group) were i.p. inoculated with $10^7$ of WNV-poly(A) or equal volume of PBS. ELISpot assay specific for IFN-γ in splenocytes was performed at 7 days post-immunization. Representative images were shown in (G) and the numbers of antibody-secreting cells (ASCs) were counted (The values represent mean $\pm$ SD in each group, each symbol represents a mouse) (H).

Data information: For statistical analysis, one-way ANOVA analysis and two-way ANOVA analysis were used for (B), (C), (F), and (E) respectively, log-rank test was performed for (D) and Mann–Whitney test was used for (H). *$P < 0.5$, **$P < 0.01$, ***$P < 0.001$,****$P < 0.0001$, ns represents not significant.

(Brien *et al*, 2007; Purtha *et al*, 2007; Welte *et al*, 2011). Non-specific stimulation of PMA and ionomycin (PMA + IONO) was used as a positive control and medium alone served as a negative control. Much higher number of IFN-γ-secreting cells were observed in the WNV-poly(A)-immunized mice compared to mock-immunized mice (600 vs 10 per $5 \times 10^5$ splenocytes; $P < 0.01$) in response to NS4B + E peptides stimulation (Fig 3H). Taken together, our data suggested that WNV-poly(A) could induce strong T-cell responses in mice after single-dose inoculation.

To further examine the durability of protective immunity induced by one shot of WNV-poly(A) vaccine, we monitored the time-course profiles of WNV-specific IgG antibody titers in C57BL/6 mice vaccinated with $10^4$ or $10^7$ PFU of WNV-poly(A) till 168 days post-immunization (Appendix Fig S8A). It showed that both titers of WNV-specific IgG antibody (Appendix Fig S8B) and neutralizing

antibody (Appendix Fig S8C) were maintained at a high level through the whole experimental period in WNV-poly(A)-vaccinated C57BL/6 mice ($n = 5$/group). At 170 days post-vaccination, the mice were challenged with WT WNV. All the mice vaccinated with either dose of WNV-poly(A) survived challenge (Appendix Fig S8D), while all the control mice vaccinated with PBS were dead within 10 days with apparent illness and viremia (Appendix Fig S8E). Collectively, the data highlighted the ability of WNV-poly(A) to induce long-term protection even with a single immunization.

## Type I IFN signaling pathway is required for WNV-poly(A) attenuation

Given that the SL and DB domains are responsible for the production of sfRNAs which modulate host innate immune responses

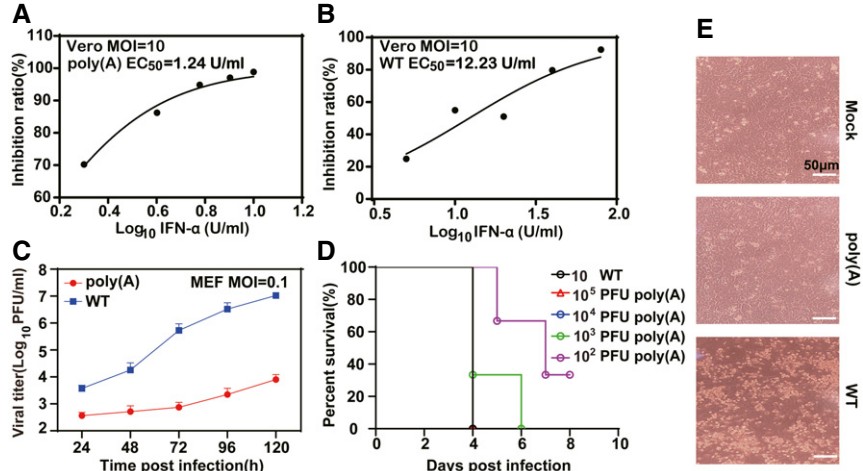

**Figure 4. Type I IFN signaling pathway is required for WNV-poly(A) attenuation.**

A, B Sensitivity of WT WNV and WNV-poly(A) viruses to interferon α (IFN-α) treatment in Vero cells. Vero cells were infected with either WT WNV or WNV-poly(A) virus at an MOI of 10 and treated with different concentrations of IFN-α. The supernatants were harvested at 48 hpi, and viral RNA copy numbers were measured by real-time RT–PCR.

C Comparison of growth kinetics between WT WNV and WNV-poly(A) viruses in MEF cells. MEF cells were infected with either WT WNV or WNV-poly(A) virus at an MOI of 0.01. Values indicate mean ± SD from two replicates of one representative experiment.

D Survival rate of IFNAR$^{-/-}$ mice infected with different dosages of WNV-poly(A) ($n = 5$).

E Cell morphology changes of MEF cells infected with WT WNV and WNV-poly(A).

Data information: Three independent assays were performed for above experiments with similar results and data of one representative data was shown. Scale bars represent 50 μm.

through type I interferon (IFN) signaling pathway, we first examined the profiles of sfRNAs in WNV-poly(A)-infected cells using Northern blot analysis. As expected, the accumulation of sfRNAs was completely abolished in WNV-poly(A) due to the absence of SL + DB domains, in contrast to an apparent sfRNA1 band observed in WT WNV (Appendix Fig S9). We then examine the sensitivity of both viruses to interferon treatment. Consistently, WNV-poly(A) displayed more sensitivity to interferon treatment than WT virus. The EC$_{50}$ of interferon treatment for inhibition of WNV-poly(A) and WT WNV RNA replication was 1.2 unit/ml (Fig 4A) and 12.2 unit/ml (Fig 4B), respectively. Additionally, the propagation of WNV-poly(A) was greatly reduced in interferon competent mouse embryonic fibroblast (MEF) cells (Fig 4C) without apparent cytopathic effect in contrast to WT WNV (Fig 4E). To further evaluate the role of type I IFN signaling pathway for WNV-poly(A) attenuation, IFNAR$^{-/-}$ mice, a C57BL/6 genetic background deficient in type I interferon (IFN) receptor, were infected with different dosage of WNV-poly(A) (Fig 4D). We observed 100% mortality of IFNAR$^{-/-}$ mice from 10$^3$ PFU WNV-poly(A) infection, in contrast to high attenuation of WNV-poly(A) in WT C57BL/6 mice without mortality with high-dosage infection (10$^7$ PFU; Fig 2). Collectively, these results suggest that the attenuation of WNV-poly(A) is in part due to its sensibility to type I interferon inhibition.

### WNV-poly(A) remains highly attenuated and immunogenic after extensive passage in cell culture

To test the genetic stability of WNV-poly(A), three independent viral stocks (a, b, and c) were subjected to 50 rounds of blind passage in Vero cells. Complete genome sequencing of all P50 viruses derived from the three independent viral stocks (a, b, and c) demonstrated that the internal poly(A) tracts were still maintained in all viral stocks even after 50 rounds of passage (Fig 5A) although some sequence variations became dominant near the junction between the poly(A) tract and the ORF stop codon (Fig 5A). Consistently, identical sized RT–PCR products were observed from P10 to P50 (Fig 5B), providing further evidence for poly(A) sequence stability. There were no visible changes in viral growth kinetics and plaque morphology between different passages of viruses (P10-, P30-, and P50-C; Fig 5C and D). Notably, WNV-poly(A)-P50 remained highly attenuated (Fig 5E). Inoculation (i.p.) with 10$^7$ PFU WNV-poly(A)-P50 did not cause any sign of diseases in mice with 100% survival rate and induced apparent seroconversion (Fig 5F) with the average neutralizing antibody titers as high as 1/160 on day 28 post-immunization (Fig 5G), completely protecting mice from challenge with 3 × 10$^7$ PFU of WT WNV (Fig 5H and I). These results show that WNV-poly(A) is safe as a LAV at least within 50 passages.

## Discussion

Targeting of 3′-UTR has been considered as an attractive strategy for the design of live-attenuated flavivirus vaccines because genetic modifications within these regions are able to attenuate the viruses without altering their antigenic specificity (Gritsun & Gould, 2006). Through deletion of different lengths of nucleotides within SL and DB domains of 3′-UTR, some LAV candidates of DENV-4 (Durbin et al, 2001) and ZIKV (Shan et al, 2017) have already been generated with good efficacy and safety. Here, we used WNV as a model to explore the feasibility of the replacement of most of 3′-UTR with

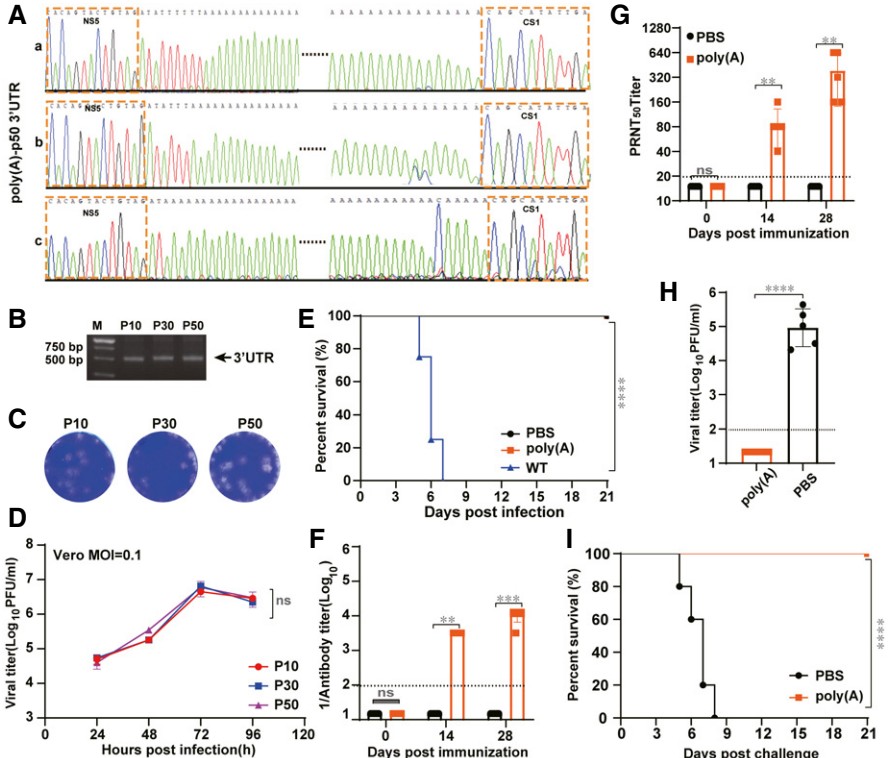

**Figure 5. WNV-poly(A) remains highly attenuated and immunogenic after extensive passage in cell culture.**

Three independent serial passages of WNV-poly(A) were performed in Vero cells.
A       Sequence chromatograms of 3′UTR in all three passaged viruses (P50-A, P50-B and P50-C).
B       RT–PCR analysis of different passages of viruses using the primer pair spanning the region from C terminus of NS5 to CS1 region of 3′UTR. The primer sequences were described in Materials and methods.
C, D   Plaque morphologies (C) and growth kinetics (D) Growth kinetics from P10, P30, and P50 using plaque assay. Three independent assays were performed and the data of one representative experiment was presented. Values indicate mean ± SD from two replicates of the experiment.
E–I    Virulence and efficacy of the passaged WNV-poly(A) (WNV-poly(A)-P50) in mice (n = 5). (E) Survival of mice infected with $10^7$ PFU of WNV-poly(A)-P50. (F) Antibody titers of mice infected with WNV-poly(A)-P50 at different days post-immunization. (G) Neutralizing antibody titers of mice infected with WNV-poly(A)-P50 at different days post-immunization (For (F) and (G), the values represent mean ± SD in each group (n = 5) and the horizontal dotted line represents the limit of detection). (H) Viremia of mice after WT WNV challenge. (The values represent mean ± SD in each group (n = 5), each symbol represents a mouse). (I) Survival of mice after WT WNV challenge.

Data information: For statistical analysis, two-way ANOVA analysis and Mann–Whitney test were used for (D) and (F), (G) respectively, one-way ANOVA analysis and log-rank test were used for (H) and (E), (I) respectively. **$P < 0.01$, ***$P < 0.001$, ****$P < 0.0001$, ns represents not significant.
Source data are available online for this figure.

internal poly(A) as a new strategy for LAV development (WNV-poly(A)). After extensive *in vitro* and *in vivo* study, we demonstrated that WNV-poly(A) is a promising vaccine candidate, which deserves further clinical development in future study.

For the first time, our study revealed that WNV is tolerated the large replacement of 3′-UTR, with the exception of 3′-sHP-SL, by internal poly(A) tract. Most importantly, the introduction of poly(A) sequence solves the problem of complete loss of viral replication capability caused by the deletion of the whole SL + DB region. Although it is still unknown regarding the underlying mechanism, at least, it provides some evidence supporting our initial speculation that maintaining a certain length of nucleotide sequence upstream from 3′-sHP-SL is required for efficient replication of flaviviruses. In addition, conservation analysis of 3′-UTRs gives us a clue that filling the SL + DB gap with polyadenylate sequence might be the optimal choice since there exist consistent adenylate-rich (A-r) spacers

between RNA secondary structure elements. As expected, WNV-poly(A) lacking the whole virulence-related SL + DB region was highly attenuated in animal model that it caused neither mortality nor any signs of illness in mice even with a high dose. Mechanistically, the attenuation of WNV-poly(A) is attributed, in part, to the decreased viral yield (Fig 1D) and increased sensitivity to type I interferon inhibition (Fig 4), as supported by the deficiency in the production of sfRNAs (Appendix Fig S9) that are involved in antagonizing type I interferon signaling (Schuessler *et al*, 2012; Chang *et al*, 2013; Manokaran *et al*, 2015; Donald *et al*, 2016). Nevertheless, the vaccine is not over-attenuated as it was able to efficiently induce long-term protective immune response (Appendix Fig S8). As a LAV, it had excellent genetic stability without virulence reversion and phenotypic alterations in grow curves and plaque morphologies even after continuous passage (~ 50 rounds) in cell culture, highlighting its high levels of safety.

Overall, we described a new internal poly(A) replacement strategy within 3′-UTR for rational design of a live-attenuated WNV vaccine. Given that most of flaviviruses share similar 3′-UTR features with heterogeneous regions (SL and DB domains) and conserved terminal 3′-sHP-3′-SL, it may represent a promising strategy to develop LAVs for other medically important flaviviruses. Nonetheless, it is equally important for us to validate its feasibility for other flaviviruses in the future work as there were studies showing that a mutation in 3′-UTR responsible for attenuation of one specific flavivirus may not necessarily be the case for other flaviviruses (Kwek *et al*, 2018).

# Materials and Methods

### Animals

Two types of mouse models used in this study, C57BL/6 mice and immunocompromised IFNAR$^{-/-}$ mice, were provided by the Animal Centre of Wuhan Institute of Virology. All the mice were cared in accordance with the recommendations of National Institutes of Health Guidelines for the Care and Use of Experimental Animals. Viral infections were conducted in an animal biosafety level 3 (ABSL-3) facility at Wuhan Institute of Virology under a protocol approved by the Laboratory Animal Ethics Committee of Wuhan Institute of Virology, Chinese Academy of Sciences (permit number: WIVA26201801).

### Cells and antibodies

BHK-21 and Vero cells were cultured in Dulbecco's modified Eagle's medium (DMEM) containing 10% fetal bovine serum (FBS), 100 units penicillin ml$^{-1}$ and 100 μg streptomycin ml$^{-1}$(PS) in 5% $CO_2$ at 37°C. BHK-21 cells were used for transfection with *in vitro* transcribed viral genomic RNA. Vero cells were used for virus propagation due to high viral titer production. Monoclonal antibody 4G2 (1:500 dilution in PBS) against flavivirus envelope protein was kindly provided by Cheng-Feng Qin (Beijing Institute of Microbiology and Epidemiology, Beijing, China). Anti-WNV capsid polyclonal antibody (1:300 dilution in PBS) was house-made by immunizing BALB/c mice with purified capsid proteins. Fluorescein isothiocyanate (FITC)-conjugated goat anti-mouse secondary antibodies (1:500 dilution in PBS) were purchased from Proteintech (China).

### Plasmid construction

The full-length infectious cDNA clone of WT WNV strain 3356 from New York City containing a T7 promotor(Shi *et al*, 2002), designated as pACYC-WT WNV, was used as the backbone to produce WNV-poly(A) plasmids and a variety of 3′UTR mutants. To construct WNV-poly(A) genome, Fragment A covering almost the whole NS5 gene and 6 nt of 5′ end of 3′ UTR (from 7,877 to 10,404 nt of WT WNV genome) was amplified by forward primer: 5′-TAA TAC GAC TCA CTA TAG TCG AAC GGA GGT TTC TCG AAC-3′ and reverse primer: 5′- TTA AAT ATC TAC AGT ACT G-3′ and inserted into pMD18T vector. Then, the resulting pMD18T-Fragment A plasmid was linearized and transcribed into RNA using T7 MEGAshortscript™ Kit (Thermo scientific). After phenol–chloroform

extraction, the poly(A) tails were added to the purified RNAs with *Escherichia coli* polyA polymerase (New England Biolabs) according to the manufacturers' instruction. Then, the resulting Fragment A-poly(A) RNA was subjected to reverse transcription polymerase chain reaction (RT–PCR) with Superscript III reverse transcription system and high-fidelity Pfu DNA polymerase, generating the cDNA fragment. At the same time, Fragment B harboring the 3′SL sequence plus 22 adenosines at 5′ end was amplified by forward primer F: 5′-AAA AAA AAA AAA AAA AAA AAA ACA GCA TAT TGA CAC CTG G-3′ and reverse primer: 5′-CGC TCT AGA CTT AAG AGA TCC TGT GTT CTC GCA CCA CCA G-3′. The long fragment named as NS5-poly(A)-3′SL was finally produced by overlap PCR using fragment A-poly(A) and fragment B as the template. Simultaneously, the genome sequence covering 5′UTR to NS5 (1–8,033, including S*pe*I restriction site) with T7 promoter sequence upstream 5′ terminus, designated as fragment L was amplified using pACYC-WT WNV as the template. Then, equal mole of S*pe*I digested long fragment NS5-poly(A)-3′SL and fragment L was *in vitro* ligated at 4°C overnight and ligation DNA was purified by phenolic-chloroform extraction for *in vitro* transcription. Deletion mutants of SL-del, DB-del, and SL + DB-del were introduced by overlap PCR, and S*pe*I and *Afl*II restriction sites were used to engineer corresponding mutant infectious clone. All the plasmids were validated by DNA sequencing analysis before the subsequent experiments. The sequencing primers are as follows: Primer-F: 5′- GAG GAG AGT GGA TGA CAA CAG AG-3′ and Primer-R: 5′-AGA TCC TGT GTT CTC GCA CCA CCA GCC-3′.

### RNA transcription, transfection, and virus rescue

All of the cDNA plasmids were subjected to sequential AflII linearization and *in vitro* transcription using mMESSENGER mMACHINE T7 Kit (Ambion, USA) following the manufacturer's instructions. WT WNV and recombinant 3′UTR mutants were rescued by electroporation of BHK-21 cells with *in vitro* transcribed viral genomic RNAs and propagated on Vero cells. To generate WNV-poly(A) virus, the supernatants from genomic RNA-electroporated cells were blindly passaged in BHK-21 cells for three rounds when relatively high viral titers were detected.

### Indirect immunofluorescence assay

At different time points post-transfection or infection, the coverslips containing transfected or infected BHK-21 cells in six-well plates were collected, washed with PBS, and fixed with cold (−20°C) 5% acetic acid in acetone for 15 min at room temperature. After washing with PBS three times, the fixed cells were incubated with 4G2 for 1 h. The cells were washed three times with PBS and then incubated with anti-mouse IgG antibody conjugated with FITC at room temperature for another hour. The cells on the coverslips were mounted with 90% glycerol and examined under a fluorescent microscope. The fluorescent images were taken at 400× magnification with a NIKON upright fluorescence microscope (Tokyo, Japan).

### Plaque assay

Virus titer and morphology were detected by monolayer plaque assay. Briefly, a series of 1:10 dilutions were prepared by diluting

15 µl virus stock with 135 µl DMEM containing 2% FBS, and 100 µl of each dilution was added into 24-well plates containing confluent BHK-21 cells (plated 1 day in advance). Infected cells were incubated at 37°C for 1 h before the overlay medium containing 2% methylcellulose was added. After 3 days of incubation, the cells were fixed in 3.7% formaldehyde and then stained with 1% crystal violet. The viral titers were calculated as plaque forming units (PFU)/ml.

### Viral growth kinetics

The virus growth kinetics of both WT WNV and WNV-poly(A) viruses were performed in Vero and MEF cells (12-well plate, $2 \times 10^5$ cells per well) at multiplicity of infection (MOI) of 0.1 and 0.01, respectively. At different time points post-infection, the culture media were collected and quantified by plaque assay in BHK-21 cells as described previously (Zhang et al, 2016; Zhang et al, 2019).

### Northern blot analysis

For WNV genome and sfRNA detection, total RNAs from infected cells were extracted with TRIzol (Invitrogen) at 4 days post-infection. About 20 µg of RNA was separated on a 1.5% agarose, 2% formaldehyde gel, followed by transfer onto Hybond-N$^+$ membranes (GE healthcare). Hybridization and visualization were performed using the DIG Northern Starter Kit I (Roche). The sequences of probe for detection were the same as described before (Zhang et al, 2020).

### IFN treatment

Vero cells seeded in 24-well plates ($1 \times 10^5$ cells per well) were infected with WT WNV or WNV-poly(A) virus at a MOI of 10 in the presence of different concentrations of IFN-α. At 48 h post-treatment, the supernatants containing different concentrations of IFN-α were harvested and viral RNA copy numbers were measured by real-time RT–PCR as described previously (Zou et al, 2009).

### Concentration and purification of viral particles

To obtain highly purified viral stocks, four 175-cm$^2$ flasks of Vero cells were infected with WT WNV or WNV-poly(A) virus at a MOI of 0.1. The supernatants reclaimed at 72 hpi were subjected to sequential centrifugation at 4°C for 10 min at 400 $g$ and for 20 min at 1,000 $g$ to remove cells and cell debris, respectively. The clarified supernatants were then mixed with polyethylene glycol 8000 (PEG8000, Sigma; to a final concentration of 8%) and incubated overnight at 4°C, followed by centrifugation at 4°C for 50 min at 10,500 $g$. The pellets were then gently re-suspended with PBS and subjected to ultracentrifugation for 1.5 h at 105,000 $g$ with 24% sucrose cushion using a SW41 rotor in an Optima MAX-XP ultracentrifuge (Beckman). The pellets were finally re-suspended with 50 µl of PBS.

### Sequence conservation analysis

Complete genome sequences of partial mosquito-borne flaviviruses were downloaded from GenBank database, corresponding to WNV ($n = 1,258$), JEV ($n = 229$), YFV ($n = 211$), ZIKV ($n = 444$), and TBEV ($n = 158$), POWV ($n = 45$), and CXFV ($n = 27$). 3′UTR sequences were derived from complete genome and aligned using MAFFT 7 (Katoh & Standley, 2013). Geneious platform was used to calculate nucleotides composition, sequences length, average identity, and number of identical sites in the 3′UTR sequences (Appendix Table S1–S7) (Kearse et al, 2012). Sequence logos of 3′UTR alignments generated by Weblogo server were used to visualize the nucleotides composition pattern and conservation(Crooks et al, 2004).

### Mouse experiments

(i) To test the pathogenesis of WNV-poly(A), cohorts of 4- to 6-week-old C57BL/6 mice ($n = 5$ per group) were injected intraperitoneally (i.p) with $10^4$–$10^7$ PFU of WNV-poly(A), $10^4$ PFU of WT WNV or PBS (negative control) in a 200 µl volume. The infected mice were monitored daily for survival, viremia, and body weight changes. To analyze viral burden and replication kinetics of WNV-poly(A) in different tissues, 6-week-old female C57BL/6 mice were inoculated subcutaneously with $10^6$ PFU of WT WNV or WNV-poly(A) in the left rear footpad, and tissues were harvested from two mice at different times post-inoculation. The viral loads in brain, spleen, left lymph node, right lymph node, and liver were determined by plaque assay. For histological analysis of brain, 4- to 6-week-old C57BL/6 mice ($n = 3$ per group) were injected i.p with $10^7$ PFU of WNV-poly(A), WT WNV, or equal volume of PBS (negative control). At 6 days post-infection (dpi), brains were harvested and made into paraffin sections followed by H&E staining.

(ii) For immunization and challenge experiments, groups of 4- to 6-week-old C57BL/6 mice ($n = 5$ per group) were immunized i.p with $10^4$–$10^7$ PFU of WNV-poly(A), PBS (negative control), or mock (untreated). At different time points, the titers of total anti-WNV IgG antibodies and neutralizing antibodies in mouse serum were examined by ELISA and PRNT assay, respectively, as described previously (Zhang et al, 2016; Zhang et al, 2019; Li et al, 2020). At 30 or 170 days post-immunization, mice were challenged i.p with $3 \times 10^7$ PFU of WT WNV. Survival and body weight changes were monitored daily till 3 weeks later. Viremia were quantified by plaque assay on day 2 post-challenge.

### Enzyme-linked immunospot (ELISPOT) assay

IFN-γ ELISpot assay was performed using the commercial mouse IFN-γ ELISpot kit (2210006, Dakewe, China) following the manufacturer's recommendations. Briefly, freshly isolated splenocytes from WNV-poly(A)-immunized mice were plated at $5 \times 10^5$ cells per well in presence of WNV peptide pool (WNV-E and WNV-NS4B, each 0.4 µg/ml) or medium alone (negative control) or 10 µg/ml of PMA and ionomycin (positive control). After incubation at 37°C, 5% $CO_2$ for 48 h, the plates were washed and incubated with biotinylated anti-IFN-γ antibody for 1 h at 37°C, followed by incubation with horseradish peroxidase (HRP)-conjugated streptavidin for 1 h at 37°C. The IFN-γ spots were developed using AEC substrate following the manufacturer's instructions. After drying, the spots were scanned and counted using ELISPOT image analysis (Bio-Sys, Karben, Germany). The average of triplicate counts of IFN-γ spot-forming cells per well was calculated by subtracting background spots from negative controls.

**The paper explained**

**Problem**

Flaviviruses include many important human pathogens which put a big burden on public health. Vaccines are powerful tools against flavivirus infection. Currently, there is no vaccine available for human use against West Nile virus.

**Results**

We have evaluated a new strategy for the development of a live-attenuated vaccine (LAV) of West Nile virus (WNV). This LAV (WNV-poly(A)) is engineered through replacement of 5′ part (SL and DB domains) of 3′UTR with an internal poly(A) in the context of WNV genome. WNV-poly(A) can propagate efficiently in Vero cells and is highly attenuated in mouse model. As a LAV, WNV-poly(A) elicits neutralizing antibodies and protects mice from wild-type (WT) WNV infection.

**Impact**

Our data demonstrate a new strategy for rational design of WNV vaccines without altering antigenic specificity.

## Statistical analysis

All data were analyzed using GraphPad Prism 8.0.2 software and expressed as mean ± standard deviation (SD). The statistical significance was assigned when $P$ values were < 0.05. Kaplan–Meier survival curves were analyzed by the log-rank test. Student's t-test was used to analyze the differences between two groups, and one-way or two-way analysis of variance (ANOVA) was utilized to analyze statistical significances among more than two groups.

## Data availability

This study includes no data deposited in external repositories.

**Expanded View** for this article is available online.

## Acknowledgements

We are grateful to the Core Facility and Technical Support (Pei Zhang, An-na Du and Juan Min), Center for Animal Experiment (Xue-fang An, Fan Zhang, He Zhao, and Li Li), and BSL-3 laboratory (Hao Tang and Jun Liu) at Wuhan Institute of Virology and Wuhan Key Laboratory of Special Pathogens and Biosafety for their helpful supports during the work. This study was supported by Beijing Shunlei Biotechnology Co. Ltd.

## Author contributions

H-QY and BZ designed and supervised the study and wrote the paper. Y-NZ, NL, and Q-YZ performed the majority of experiments and analyzed the data. JL, S-LZ, LG, X-YZ, FY, H-QZ, XDL, C-LD, and S-PY contributed specific experiments and data analysis. P-YS and Z-MY provided some materials for experiments. All authors read and approved the contents of the manuscript.

## Conflict of interest

The authors declare no competing interests. There is a pending patent application related to the study. S.-L.Z., L.G., and S.-P.Y. are employees of Beijing Shunlei Biotechnology Co. Ltd. which supported this study.

## For more information

WNV homepage of CDC: www.cdc.gov/westnile/index.html

WNV introduction of NIAID: www.niaid.nih.gov/diseases-conditions/west-nile-virus

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
