## [Review Process File · EMBO Molecular Medicine]

Rational design of West Nile virus vaccine through large replacement of 3' UTR with internal poly(A)

Ya-Nan Zhang, Na Li, Qiu-Yan Zhang, Jing Liu, Shun-Li Zhan, Lei Gao, Xiang-Yue Zeng, Fang Yu, Hong-Qing Zhang, Xiao-Dan Li, Cheng-Lin Deng, Pei-Yong Shi, Zhi-Ming Yuan, Shao-Peng Yuan, Han-Qing Ye, Bo Zhang

DOI: [10.15252/emmm.202114108](https://doi.org/10.15252/emmm.202114108)

Corresponding author: Bo Zhang (zhangbo@wh.iov.cn)

Review Timeline:

Submission Date:	9th Feb 21
Editorial Decision:	11th Mar 21
Revision Received:	8th Jun 21
Editorial Decision:	25th Jun 21
Revision Received:	30th Jun 21
Accepted:	9th Jul 21

Editor: Zeljko Durdevic

Transaction Report:

11th Mar 2021

Dear Prof. Zhang,

Thank you for the submission of your manuscript to EMBO Molecular Medicine. We have now received feedback from the three reviewers who agreed to evaluate your manuscript. As you will see from the reports below, the referees acknowledge the interest of the study but also raise serious and partially overlapping concerns that should be addressed in a major revision. Our cross-commenting session made it clear that: a) The universality of the approach is not supported by the data and should be addressed either by testing other flaviviruses or by changing the title of the manuscript, b) Experiments should be performed with the plaque purified virus or a virus derived from an infectious clone with defined poly(A) length, and c) Analysis of immunity in wild-type mice generated by UV-inactivated virus would strengthen the conclusion that the virus is acting as an attenuated, rather than inactivated, antigen.

Addressing the reviewers' concerns in full will be necessary for further considering the manuscript in our journal, and acceptance of the manuscript will entail a second round of review. EMBO Molecular Medicine encourages a single round of revision only and therefore, acceptance or rejection of the manuscript will depend on the completeness of your responses included in the next, final version of the manuscript. For this reason, and to save you from any frustrations in the end, I would strongly advise against returning an incomplete revision.

We realize that the current situation is exceptional on the account of the COVID-19/SARS-CoV-2 pandemic. Therefore, please let us know if you need more than three months to revise the manuscript.

I look forward to receiving your revised manuscript.

Yours sincerely,

Zeljko Durdevic

***** Reviewer's comments *****

Referee #1 (Comments on Novelty/Model System for Author):

The manuscript is well and clearly written. The methodology is adequate and somehow innovative.

The results are a significant step forward in flavivirus vaccinology and are also interesting for non-specialist.

Referee #1 (Remarks for Author):

Manuscript of Zhang et al. describes the production and characterization of VNO-poly A mutants in which the 5' portion (SL and BD domains) of the 3' UTR is replaced by a poly A tract. The mutant replicates in cell culture and is stable after serial passages. Inoculation of a single dose of the mutant into susceptible mice induces a long-lasting protective response against a lethal challenge from the WT virus, making it an attractive candidate vaccine.

The manuscript is well and clearly written. The methodology is adequate and somehow innovative. The results are a significant step forward in flavivirus vaccinology and are also interesting for non-specialist.

Comments:

As only WNV mutants have been tested, and although it is possible that the same strategy will work for other flaviviruses, as stated in the text (line 302), this should be specifically addressed and, thus, title should be changed so, instead of "Rational design of Flaviviruses...." It will be more accurate to change flaviviruses by WNV.

How do the authors explain that the mutant does not induce viremia in the inoculated mice, but that the animals produce neutralizing antibodies and are protected?

Line 77, it should be indicated that, as for other flaviviruses, there are already licensed vaccines for WNV.

The quality of Fig. S4 is not good enough.

Although the use of diverse cell lines in the different experiments is adequate, a short explanation for doing so should be included for readers not familiarized with the topic, and the same for the inoculation route (i.p., sc) use in the different experiments.

Panels in the text are in capital letters, while in the figures are in low case letters.

Referee #2 (Comments on Novelty/Model System for Author):

Additional work is needed to test the replication of the virus in mosquitoes

Referee #2 (Remarks for Author):

Zhang and colleagues report the rational design of an experimental live attenuated West Nile virus (WNV) vaccine by replacing a nonessential ~520-nt region (called here: SL and DB) within the 3' NCR of the WNV genome with an internal poly(A) tract. Remarkably, these viruses replicated remarkably well in cell cultures (as long as the cells were devoid of innate immunity) and could induce protective immune responses after injection into mice. This study is premised on and derivative of two prior studies (dating back twenty years) showing that small deletions within the 3' NCR of other flaviviruses leads to attenuation. The key novelty is that the authors have substituted a poly(A) tract for larger deleted regions, which was apparently key to allow replication. They pitch this as a "universal" approach to flavivirus vaccine design but do not demonstrate the broad applicability of this approach to other viruses. Other concerns include: lack of rigorous genetic analysis, equivocal data that WNV-poly(A) actually replicates in mice, and cursory analysis of other flaviviruses and host species. Specific issues for consideration:

Major concerns:

1. In lines 139-153 and Fig. S4, the authors argue that functional structural elements within flavivirus 3' NCRs are separated by A-rich (A-r) regions. Lines 142-146 indicate that these distributions are statistically significant. However, the statistical comparisons are unclear, as written. Are the unstructured "spacer" regions specifically enriched for adenine vs. the structured regions? And is this a conserved feature of flaviviruses? Is this a feature only of mosquito-borne flaviviruses? Or do these observations extend to tick-borne flaviviruses, arthropod-specific flaviviruses, and/or flaviviruses with no known vector? In short, this line of argument provides the theoretical basis for the approach and should be explained more clearly and with increased rigor.
2. The WNV-poly(A) was selected by serial passage, sequenced, then used for experiments. However, no effort was described to plaque-purify the resultant virus or to reconstruct the winning WNV-poly(A) genome, so it is entirely possible that the replication-competent virus is a mixed population, at least some of which are capable of forming plaques.
3. What other sequence change(s) were detected in the passaged pool?
4. What were the specific infectivities of the WT vs. WNV-poly(A) virus particles (i.e., number of infectious units per virus particle)?
5. Fig. 2 provides very little evidence that WNV-poly(A) actually replicates in mice. Viremia was only detected on day 1 (Fig. 2C), and only in mice infected with the highest dose (1E7 PFU), suggesting that the authors are simply reisolating the input virus. Similarly, with 1E6 PFU input, only 1 of 5 mice showed detectable WNV-poly(A) in spleen at day 4. Finally, why are there error bars on the WT day 2 (Fig. 2e), but not in the other samples?
6. The authors have shown that WNV-poly(A) does not cause damage to the brain (Fig. 2d); however, since it is unclear whether the virus really replicating *in vivo*, it is premature to say that the virus is not neuroinvasive (lines 209-210). I would also like to know whether WNV-poly(A) is neurovirulent if mice are infected intracranially.
7. The induction of a protective antibody response suggests that WNV-poly(A) does indeed infect mice. However, this is rather indirect, as it could also be that the input virus particles were sufficient to induce this response on their own, especially since the specific infectivity was never determined. One simple experiment to demonstrate this would be to inactivate the virus particles with UV or H₂O₂ (which is more gentle on antigens but kills the RNA genome). If the Ab response is reduced, this would suggest that viral replication is needed; if the Ab response is not reduced, this would suggest that the input virus particles are capable of inducing this response on their own, in a replication-independent manner.
8. Fig. 5a does not fully support the statement "that the internal poly(A) tracts were stably maintained" as claimed in lines 278-279. First, this Figure does not show the lengths of the poly(A) tracts. Were these lengths maintained? And the data do show that some sequence variations became dominant near the junctions between the poly(A) tract and flanking viral genome. Over time, one could imagine that new RNA structures will arise, some of which may phenocopy the functions of the missing SL and DB regions.
9. Since the WNV-poly(A) will presumably be used as a vaccine, and WNV is transmitted from humans to mosquitoes, it is important to examine whether WNV-poly(A) is capable of replication in mosquitoes (or, at a minimum, relevant mosquito cell lines). One could imagine that a vaccinated person will pass the WNV-poly(A) virus to mosquitoes, which will amplify the virus and select for additional variants. Oddly, the authors describe methods for growing C6/36 (*Aedes albopictus*) cells, but they do not present any data (in any case, this is the wrong mosquito line to use for WNV).
10. Lines 279-299 are inaccurate and an overstatement. This study did not examine whether poly(A) replacement of the 3' SL and DB regions is a universal strategy for vaccine development, since only WNV was examined.

Minor concerns:

11. Lines 72-73. Please follow the International Committee on the Taxonomy of Viruses (ICTV)

nomenclature. Virus names should not be capitalized unless they contain a proper noun (e.g., named after a person or place).

12. Fig. 2D is too small to be legible.

13. Line 168 "did be" should be "was."

14. Line 200 "confirmed" should be "confirm."

15. The Bibliography contains typos and incomplete references (e.g., lines 608-609).

16. Line 686 "adverse" should be "reverse."

Referee #3 (Remarks for Author):

Synopsis: The authors of this manuscript hypothesize that a replication competent, but attenuated flavivirus can be obtained via deletion of the majority of the 3' UTR and replacement of this region with a poly(A) tract. The hypothesis is based on previous data demonstrating: 1) attenuation of various flaviviruses with partial 3' UTR deletions; 2) rescue of non-replicating 3' UTR-deletion viruses by duplication of existing sequences, suggesting a minimum 3'UTR length required for viral replication; 3) the presence of regions of high A content in several regions of the UTRs of multiple flaviviruses. As previously shown, the authors were unable to recover virus following transfection of WNV RNA from a clone with a deletion of all the 3'UTR, except the 3' stem-loop. Addition of multiple A's in place of the deletion to this clone via polyA polymerase and fusion PCR resulted in recovery of virus in BHK-21 cells transfected with the viral RNA ("WNV-poly(A)"). The authors then convincingly show that this virus is highly attenuated in C57Bl/6 mice, and that attenuation is stable through extensive cell culture passage in Vero cells. Further, they show that infection with WNV-poly(A) elicits a robust immune response that is protective against infection with wild-type WNV. Finally, they show that the 3'UTR-derived subgenomic flavivirus RNA (sfRNA) is absent from the WNV-poly(A) virus. Because the sfRNA is associated with modulation of the host interferon response, the authors hypothesize that the virus is sensitive to interferon. They show this experimentally in culture by addition of exogenous interferon, as well as in vivo, showing greater lethality of the WNV-poly(A) virus in IFNAR^{-/-} mice.

Comments/ concerns:

1. In general, the data support the authors conclusions. However, they suggest several times that this strategy can be applied to all mosquito-borne flaviviruses. This is certainly a reasonable hypothesis, but it has not been tested, since data is only shown for WNV. Notably, small (30 nt) deletions in the 3' UTR of multiple dengue virus serotypes decrease expression of the sfRNA for all these viruses, but the extent of attenuation differs for the different serotypes. The use of polyA substitution for the 3'UTR should therefore be the focus of a speculative paragraph in the discussion.

2. It would strengthen the manuscript is the authors transfected RNAs with defined poly(A) lengths (rather than a pool of multiple lengths) to define what the required length of insert is.

3. What is the red arrow in Fig 2D? The text refers to black, yellow, and blue arrows, but not red. It seems to be an apoptotic cell, although examination of this figure requires significant zooming in on a computer screen.

4. The T cell data in figures 3g and 3h are confusing. The PMA+ionomycin treatment should induce all T-cells capable of secreting IFN γ to do so. The results in 3h suggest that the majority of T-cells in the spleens of vaccinated mice are responding to WNV peptides- this seems high? This experiment should also include an irrelevant (non-WNV) peptide control.

Minor points:

5. The manuscript should be looked over for standard English grammar and usage (mostly missing indefinite/ definite articles). Line 168: "did be" should be changed to "was" or "is."

11th Mar 2021

Dear Prof. Zhang,

Thank you for the submission of your manuscript to EMBO Molecular Medicine. We have now received feedback from the three reviewers who agreed to evaluate your manuscript. As you will see from the reports below, the referees acknowledge the interest of the study but also raise serious and partially overlapping concerns that should be addressed in a major revision. Our cross-commenting session made it clear that: a) The universality of the approach is not supported by the data and should be addressed either by testing other flaviviruses or by changing the title of the manuscript, b) Experiments should be performed with the plaque purified virus or a virus derived from an infectious clone with defined poly(A) length, and c) Analysis of immunity in wild-type mice generated by UV-inactivated virus would strengthen the conclusion that the virus is acting as an attenuated, rather than inactivated, antigen.

Addressing the reviewers' concerns in full will be necessary for further considering the manuscript in our journal, and acceptance of the manuscript will entail a second round of review. EMBO Molecular Medicine encourages a single round of revision only and therefore, acceptance or rejection of the manuscript will depend on the completeness of your responses included in the next, final version of the manuscript. For this reason, and to save you from any frustrations in the end, I would strongly advise against returning an incomplete revision.

We realize that the current situation is exceptional on the account of the COVID-19/SARS-CoV-2 pandemic. Therefore, please let us know if you need more than three months to revise the manuscript.

I look forward to receiving your revised manuscript.

Yours sincerely,

Zeljko Durdevic

Zeljko Durdevic

Editor

EMBO Molecular Medicine

Response: Many thanks for letting us resubmit our revised manuscript. We have addressed the questions point-by-point in our response letter. We also performed additional experiments according suggestions from reviewers and added additional data in the revised manuscript.

a) We have changed the title of the manuscript by replacing "flaviviruses" with "WNV".

b) In our original study, we found that internal poly(A) is around 130 nt after three rounds of passage which was also described in our original manuscript. Following the suggestions of reviewers, several individual plaque purified viruses were selected, and similar length of internal poly(A) sequence of around 130 nt were observed as pooled viruses from P3. We arbitrarily selected one plaque purified virus for virulence and immunization study, and similar data were obtained as pooled viruses. All these data were presented in our revised manuscript (Appendix Figure S7.)and below for your reference.

Appendix Figure S7. Plaque purified WNV-poly(A) for virulence and immunization study in C57BL/6 mice.

c) As suggested, we first inactivated WNV-poly(A) and immunized mice with 10⁵ PFU of UV-inactivated WNV-poly(A) or WNV-poly(A). Our results directly demonstrated that UV-inactivated virus could not induce antibody reactivity in contrast to seroconversion against WNV of WNV-poly(A). The data were presented below and in our revised manuscript as Appendix Figure S6C.

Total anti-WNV IgG antibody of WNV-poly(A) and UV- inactivated WNV-poly(A).

***** Reviewer's comments *****

Referee #1 (Comments on Novelty/Model System for Author):

The manuscript is well and clearly written. The methodology is adequate and somehow innovative. The results are a significant step forward in flavivirus vaccinology and are also interesting for non-specialist.

Referee #1 (Remarks for Author):

Manuscript of Zhang et al. describes the production and characterization of WNV-poly A mutants in which the 5' portion (SL and BD domains) of the 3' UTR is replaced by a poly A tract. The mutant replicates in cell culture and is stable after serial passages. Inoculation of a single dose of the mutant into susceptible mice induces a long-lasting protective response against a lethal challenge from the WT virus, making it an attractive candidate vaccine.

The manuscript is well and clearly written. The methodology is adequate and somehow innovative. The results are a significant step forward in flavivirus vaccinology and are also interesting for non-specialist.

Comments:

As only WNV mutants have been tested, and although it is possible that the same strategy will work for other flaviviruses, as stated in the text (line 302), this should be specifically addressed and, thus, title should be changed so, instead of "Rational design of Flaviviruses...." It will be more accurate to change flaviviruses by WNV.

Response: Many thanks for the comment. We have changed the title of this manuscript by replacing "flaviviruses" with "WNV".

How do the authors explain that the mutant does not induce viremia in the inoculated mice, but that the animals produce neutralizing antibodies and are protected?

Response: Many thanks for your comment. In contrast to WT WNV, we found that WNV-poly(A) is sensitive to interferon treatment in Vero cells, and could not replicate efficiently in MEF cells (Fig 4). Thus, we speculate the reason that WNV-poly(A) could not induce viremia is because it could not replicate efficiently in normal mice caused by its sensitivity to interferon response. Nonetheless, WNV-poly(A) could still infect mice with limited replication that is enough to stimulate immune responses.

Line 77, it should be indicated that, as for other flaviviruses, there are already licensed vaccines for WNV.

Response: Many thanks for your comment. Currently, there is no licensed WNV vaccine for human use although the U.S. Department of Agriculture licensed a DNA vaccine to prevent WNV in horses in 2005, and since then, at least four other types of WNV vaccines have been approved for use in horses. (West Nile Virus Vaccines, <https://www.niaid.nih.gov/diseases-conditions/wnv-vaccines>). At the same time, some technologies were transferred into clinical testing, but these approaches have not yet led to a licensed product for human use. (Hum Vaccin Immunother. 2019;15(10):2337-2342.)

To avoid misleading, we changed "Despite the availability of licensed vaccines for" to "Despite the availability of licensed vaccines for human use such as" line 99.

The quality of Fig. S4 is not good enough.

Response: We have replaced a new figure with high resolution for Fig. S4.

Although the use of diverse cell lines in the different experiments is adequate, a short explanation for doing so should be included for readers not familiarized with the topic, and the same for the inoculation route (i.p., sc) use in the different experiments.

Panels in the text are in capital letters, while in the figures are in low case letters.

Response: Many thanks for the comments.

- 1) In our lab, BHK-21 is more efficient to produce recombinant virus through transfection with in vitro transcribed WNV RNAs than other cells such as Vero cell, that is why we used BHK-21 cells for transfection. In contrast, WNV could replicate more efficiently with higher viral titers in Vero cell than in BHK-21 cells, thus we usually use Vero cells to propagate West Nile virus. We have added a short explanation in Materials and Methods (Cells and antibodies section) in line 477-479 in our revised manuscript.
- 2) For WNV vaccine study, immunization through i.p. or s.c. route is widely accepted. In this study, we used i.p. route for immunization (PMID: 23221549, J Virol. 2013 Feb;87(4):1926-36.; PMID: 23544010, PLoS Pathog. 2013 Feb;9(2):e1003168.). We infected mice through s.c. at footpad, which is specially used to examine the spread of WNV in mice (PMID: 23544010, PLoS Pathog. 2013 Feb;9(2):e1003168.). We added reference publication at line 274-275 in our revised manuscript.
- 3) We changed all low case letters to capital letters in the figures.

Referee #2 (Comments on Novelty/Model System for Author):

Additional work is needed to test the replication of the virus in mosquitoes

Response: Many thanks for your suggestions. As we lack the facility for mosquito experiments, we cannot test the replication of the virus in mosquitoes. We hope you will understand our present situation. We performed viral replication in mosquito cell line, which demonstrated that WNV-poly(A) could replicate in mosquito cell line but less efficiently than WT WNV. The data were presented below for questions 9.

At the same time, it has been reported that SL and DB regions contribute both sfRNA production and viral transmission in mosquitoes (PMID:27581979, J Virol. 2016 Oct 28;90(22):10145-10159; PMID:32581095, J Virol. 2020 Aug 31;94(18):e00343-20.; PMID:31488709, Proc Natl Acad Sci U S A. 2019 Sep 17;116(38):19136-19144). We have demonstrated that WNV-poly(A) could not produce sfRNA with poly(A) tracts replacement of both SL and DB regions, we speculate that WNV-poly(A) may significantly decrease viral infection and transmission rates in mosquitoes.

Referee #2 (Remarks for Author):

Zhang and colleagues report the rational design of an experimental live attenuated West Nile virus (WNV) vaccine by replacing a nonessential ≈520-nt region (called here: SL and DB) within the 3' NCR of the WNV genome with an internal poly(A) tract. Remarkably, these viruses replicated remarkably well in cell cultures (as long as the cells were devoid of innate immunity) and could induce protective immune

responses after injection into mice. This study is premised on and derivative of two prior studies (dating back twenty years) showing that small deletions within the 3' NCR of other flaviviruses leads to attenuation. The key novelty is that the authors have substituted a poly(A) tract for larger deleted regions, which was apparently key to allow replication. They pitch this as a "universal" approach to flavivirus vaccine design but do not demonstrate the broad applicability of this approach to other viruses. Other concerns include: lack of rigorous genetic analysis, equivocal data that WNV-poly(A) actually replicates in mice, and cursory analysis of other flaviviruses and host species. Specific issues for consideration:

Response: Many thanks for your comments. Since we did not test our method in other flavivirus, we changed the title of the manuscript by replacing "flaviviruses" with "WNV" and tuned down some statement in the text of the manuscript. As suggested, we performed additional experiments and added new data in our revised manuscript.

Major concerns:

1. In lines 139-153 and Fig. S4, the authors argue that functional structural elements within flavivirus 3' NCRs are separated by A-rich (A-r) regions. Lines 142-146 indicate that these distributions are statistically significant. However, the statistical comparisons are unclear, as written. Are the unstructured "spacer" regions specifically enriched for adenine vs. the structured regions? And is this a conserved feature of flaviviruses? Is this a feature only of mosquito-borne flaviviruses? Or do these observations extend to tick-borne flaviviruses, arthropod-specific flaviviruses, and/or flaviviruses with no known vector? In short, this line of argument provides the theoretical basis for the approach and should be explained more clearly and with increased rigor.

Response: Thank you for your comments and suggestions. As Lines 142-146 mentioned in original manuscript, the statistical comparisons were performed to compare proportion of adenosine nucleotides with that of guanine nucleotides, cytidine nucleotides, uracil nucleotides in A-r segments to prove that there are indeed A-rich (A-r) segments in 3' UTRs.

As suggested, in order to prove whether A-rich (A-r) region is the conserved feature of flaviviruses, besides WNV, JEV, YFV and ZIKV, we expanded the analysis scope and added TBEV (n=158), POWV (n=45) and CXFV (n=27), since the number of NKFV was less than 10, it was not analyzed. The sequences alignment reveals that A-r segments composed of 6-28 adenosine nucleotides exist in 3'-UTRs of all above analysed viruses. And for further analysis, we found that there is no obvious difference of A-r spacers in structured or unstructured region of 3' UTR.

We have revised the sentence in line 171-177 in revised manuscript to read "We expanded the analysis to more MBFVs, TBFVs and ISFVs, since the number of NKFV was less than 10, it was not analyzed. The sequences alignment reveals that A-r segments with significantly higher adenosine ribonucleotides composition ($p < 0.05$) exist in 3'-UTRs of WNV, JEV, YFV, ZIKV, TBEV, POWV and CXFV (Appendix Figure S4 and Table S1-S7), and the A-r segments composed of 6-28 adenosine nucleotides were widely distributed in structured and unstructured domains of 3'-UTRs."

2. The WNV-poly(A) was selected by serial passage, sequenced, then used for experiments. However, no effort was described to plaque-purify the resultant virus or to reconstruct the winning WNV-poly(A) genome, so it is entirely possible that the replication-competent virus is a mixed population, at least some of which are capable of forming plaques.

Response: Many thanks for your comment. In fact, we randomly picked some plaques from Passage 3 pooled viruses and amplified in cell culture, we found that the length of poly(A) tracts of plaque purified viruses is similar to that of Passage 3 pooled viruses. The sequencing and RT-PCR data were presented as Fig. 1C and D.

3. What other sequence change(s) were detected in the passaged pool?

Response: We sequenced the complete genome of three passaged P50 viruses. Besides the engineered poly(A) tracts within 3'UTR that were stably maintained during passage, various cell adaptative mutations were found mostly in nonstructural proteins region. These mutations seemly occurred randomly since there are no identical mutations within all three independently passaged viruses. Such randomly generated mutations have little possibility to increase viral virulence, as supported by the virulence assay of passaged viruses in Fig.5.

WNV-poly(A)-P50			
	a	b	c
NS1	—	V54V GTG→GTA K170N AAG→AAT(Mix)	—
NS2A	P8P CCT→CCA	T20T ACC→ACT	A30V GCC→GTC
NS2B	I124L ATA→CTA	—	I124L ATA→CTA
NS3	G114G GGG→GGA T356P ACC→CCC	—	T356P ACC→CCC G445R GGA→AGA(Mix)
NS4A	Q73H CAG→CAT	A55V GCC→GTC K124R AAG→AGG	L44M CTG→ATG F92L TTT→CTT V135M GTG→ATG
NS4B	T241I ACA→ATA I245T ATA→ACA(Mix)	S71S TCA→TCC T241I ACA→ATA I245T ATA→ACA	C120F TGC→TTC
NS5	P136S CCT→TCT L481L CTT→CTC	H173R CAC→CGC L481L CTT→CTC L640L CTC→CTT	G106G GGC→GGT N234N AAT→AAC L481L CTT→CTC C670W TGT→TGG(Mix) A871A GCT→GCC
E	P360P CCT→CCC	E390G GAA→GGA	—

4. What were the specific infectivities of the WT vs. WNV-poly(A) virus particles (i.e., number of infectious units per virus particle)?

Response: Many thanks for your comment. We loaded equal amounts (10^6 PFU) of the two viruses for Western Blotting assay using anti-Envelope polyclonal antibody to quantify the levels of viral envelope protein, and similar amounts of viral envelope proteins were observed. Our results indicated that specific infectivities were similar for both WNV-poly(A) and WT WNV. One of representative data was also presented below and in our revised manuscript as Appendix Figure S6A.

5. Fig. 2 provides very little evidence that WNV-poly(A) actually replicates in mice. Viremia was only detected on day 1 (Fig. 2C), and only in mice infected with the highest dose ($1E7$ PFU), suggesting that the authors are simply reisolating the input virus. Similarly, with $1E6$ PFU input, only 1 of 5 mice showed detectable WNV-poly(A) in spleen at day 4. Finally, why are there error bars on the WT day 2 (Fig. 2e), but not in the other samples?

Response: Many thanks for your comments. We do agree that Fig. 2 provides little evidence that WNV-poly(A) actually replicates in mice, and the viremia detected on day 1 post infection in mice infected with the highest dose are simply reisolating the input virus. However, we would like to stress that the data presented in Fig. 2 were used to demonstrate that WNV-poly(A) is highly attenuated in mice which is safe as a vaccine candidate.

We do believe that WNV-poly(A) could replicate in mice, albeit with much lower efficiency comparing with that of WT WNV, supported by several experiments:

1) WNV-poly(A) could replicate in mouse embryonic fibroblast (MEF) cells although replication efficiency is lower than that of WT WNV (Fig. 4C).

2) WNV-poly(A) is sensitive to IFN treatment (Fig 4A and B) in contrast to WT WNV, and is lethal for A129 mice, which indicated that WNV-poly(A) could infect and replicate in A129 mice but could not replicate efficiently in normally mice such as C57BL/6.

3) We performed the experiments as suggested that we immunized mice with 10^5 PFU of WNV-poly(A) and 10^5 PFU of UV-inactivated WNV-poly(A). WNV-poly(A) could induce antibody response in contrast to no detectable antibody activity of UV-inactivated WNV-poly(A), which indirectly indicated that WNV-poly(A) could replicate in normal mice. The new data were presented as Appendix Figure S6C in our revised manuscript.

We added error bars on the WT day 2 (Fig. 2E) and replaced original figure with new figure. We are sorry for this mistake.

6. The authors have shown that WNV-poly(A) does not cause damage to the brain (Fig. 2d); however, since it is unclear whether the virus really replicating in vivo, it is premature to say that the virus is not neuroinvasive (lines 209-210). I would also like to know whether WNV-poly(A) is neurovirulent if mice are infected intracranially.

Response: Many thanks for your comments. As we answered question 5 above, we do believe that WNV-poly(A) could replicate in mice although the viral replication efficiency is low. In addition, from Fig. 2D and 2E results, we do believe that it is reasonable to say that the virus is not neuroinvasive.

As reviewer suggested, we infected 4-week-old ICR mice intracranially with different dosages of WNV-poly(A), and found that all mice succumbed within 7 days when infected with more than 10^6 PFU WNV-poly(A) although average survival time (AST) of WNV-poly(A) is one day longer than that of

WT-WNV, which indicated that WNV-poly(A) is still neurovirulent for mice. Since our strategy for attenuation targeting 3'-UTR of WNV, it is reasonable that neurovirulent for mice of WNV-poly(A) is not attenuated because the envelope of WNV-poly(A) is exactly same as WT-WNV. The new data were presented in our revised manuscript as Appendix Figure S6B.

7. The induction of a protective antibody response suggests that WNV-poly(A) does indeed infect mice. However, this is rather indirect, as it could also be that the input virus particles were sufficient to induce this response on their own, especially since the specific infectivity was never determined. One simple experiment to demonstrate this would be to inactivate the virus particles with UV or H₂O₂ (which is more gentle on antigens but kills the RNA genome). If the Ab response is reduced, this would suggest that viral replication is needed; if the Ab response is not reduced, this would suggest that the input virus particles are capable of inducing this response on their own, in a replication-independent manner.

Response: Many thanks for your suggestions. We first inactivated WNV-poly(A) with UV for 30 minutes and the inactivation of viruses was confirmed by plaque assay. 10⁵ PFU WNV-poly(A) and UV-inactivated WNV-poly(A) were used to immunize C57/B6 mice, respectively. Total IgG antibody titers against WNV in the sera from immunized mice were measured on day 14 and 28 using ELISA assay. Seroconversion against WNV were observed in WNV-poly(A) immunized mice in contrast to UV-inactivated WNV-poly(A) immunized mice, which suggested that WNV-poly(A) indeed infected mice to induce antibody response. The new data were presented above in the response to editor and as Appendix Figure S6C in our revised manuscript.

8. Fig. 5a does not fully support the statement "that the internal poly(A) tracts were stably maintained" as claimed in lines 278-279. First, this Figure does not show the lengths of the poly(A) tracts. Were these lengths maintained? And the data do show that some sequence variations became dominant near the junctions between the poly(A) tract and flanking viral genome. Over time, one could imagine that new RNA structures will arise, some of which may phenocopy the functions of the missing SL and DB regions.

Response: Many thanks for your comments. We performed RT-PCR assay to detect the presence of poly(A) tract using primer pairs spanning from C-terminus of NS5 to CS1 region of 3'UTR for different passages of viruses and identical sized RT-PCR products were obtained which indicated the length of poly(A) tracts were maintained after serial passage. The new data were presented as Fig. 5B. We do agree that some sequence variations were observed from different passaged viruses near the junctions between the poly(A) tract and flanking viral genome after extensive passages (50 passages). From the sequencing data (Fig. 5A), and RT-PCR results (Fig. 5B) we did believe that poly(A) tracts were maintained, and it is very unlikely that new RNA structures arise.

9. Since the WNV-poly(A) will presumably be used as a vaccine, and WNV is transmitted from humans to mosquitoes, it is important to examine whether WNV-poly(A) is capable of replication in mosquitoes (or, at a minimum, relevant mosquito cell lines). One could imagine that a vaccinated person will pass the WNV-poly(A) virus to mosquitoes, which will amplify the virus and select for additional variants. Oddly, the authors describe methods for growing C6/36 (*Aedes albopictus*) cells, but they do not present any data (in any case, this is the wrong mosquito line to use for WNV).

Response: Many thanks for your comments.

1) Among WNV transmission cycle, birds such as crow, and humans are amplifying host and dead-end host, respectively. Namely, mosquitoes become infected when they feed on infected birds. Infected mosquitoes then spread WNV to people and other animals by biting them. (<https://www.cdc.gov/westnile/transmission/index.html>). In this case, WNV could not be transmitted from humans to mosquitoes. It is thus very unlikely that WNV-poly(A) is transmitted from humans to mosquitoes when WNV-poly(A) is used as a vaccine.

Additionally, as we have discussed above that SL and DB regions of 3UTR contribute to both sfRNA production and viral transmission in mosquitoes (PMID:27581979, J Virol. 2016 Oct 28;90(22):10145-10159; PMID:32581095, J Virol. 2020 Aug 31;94(18):e00343-20.; PMID:31488709, Proc Natl Acad Sci U S A. 2019 Sep 17;116(38):19136-19144). Since WNV-poly(A) could not produce sfRNA due to the replacement of SL and DB regions with poly(A) tract, we speculate that WNV-poly(A) will significantly decrease viral infection and transmission rates in mosquitoes.

2) Although we do not think WNV-poly(A) could transmit from humans to mosquitoes, we still examined the replication of WNV-poly(A) in mosquito cells and data were presented below. As we do not have the facilities to perform viral replication assays in mosquitoes, we only tested WNV-poly(A) replication in mosquito cells. We hope you will understand our present situation.

3) In our original manuscript, we made mistake to describe methods for growing C6/36 mosquito cell line. We are sorry for this mistake.

4) We used C6/36 instead of other mosquito cell lines for WNV-poly(A) replication because of two reasons.

a) C6/36 is the only mosquito cell line we have at hand in our lab. b) Mosquito species in which West Nile virus has been detected include *Aedes* species besides *Culex* species. <https://www.cdc.gov/westnile/resources/pdfs/MosquitoSpecies1999-2016.pdf>. C6/36 cell line is derived from *Aedes albopictus* mosquitoes. In spite of lacking *in vivo* data currently, we do believe the results of WNV-poly(A) in C6/36 mosquito cells could mirror *in vivo* results at some extent.

10. Lines 279-299 are inaccurate and an overstatement. This study did not examine whether poly(A) replacement of the 3' SL and DB regions is a universal strategy for vaccine development, since only WNV was examined.

Response: Many thanks for your comment. We have rephrased the sentence and tuned down the statement. Additionally, we also changed the title of the manuscript by replacing "flaviviruses" with "WNV".

Minor concerns:

11. Lines 72-73. Please follow the International Committee on the Taxonomy of Viruses (ICTV) nomenclature. Virus names should not be capitalized unless they contain a proper noun (e.g., named after a person or place).

Response: Corrected as suggested.

12. Fig. 2D is too small to be legible.

Response: Fig. 2D was re-organized in our revised manuscript.

13. Line 168 "did be" should be "was."

Response: Corrected.

14. Line 200 "confirmed" should be "confirm."

Response: Corrected.

15. The Bibliography contains typos and incomplete references (e.g., lines 608-609).

Response: Corrected.

16. Line 686 "adverse" should be "reverse."

Response: Corrected.

Referee #3 (Remarks for Author):

Synopsis: The authors of this manuscript hypothesize that a replication competent, but attenuated flavivirus can be obtained via deletion of the majority of the 3' UTR and replacement of this region with a poly(A) tract. The hypothesis is based on previous data demonstrating: 1) attenuation of various flaviviruses with partial 3' UTR deletions; 2) rescue of non-replicating 3' UTR-deletion viruses by duplication of existing sequences, suggesting a minimum 3'UTR length required for viral replication; 3) the presence of regions of high A content in several regions of the UTRs of multiple flaviviruses. As previously shown, the authors were unable to recover virus following transfection of WNV RNA from a clone with a deletion of all the 3'UTR, except the 3' stem-loop. Addition of multiple A's in place of the deletion to this clone via polyA polymerase and fusion PCR resulted in recovery of virus in BHK-21 cells transfected with the viral RNA ("WNV-poly(A)"). The authors then convincingly show that this virus is highly attenuated in C57Bl/6 mice, and that attenuation is stable through extensive cell culture passage in Vero cells. Further, they show that infection with WNV-poly(A) elicits a robust immune response that is protective against infection with wild-type WNV. Finally, they show that the 3'UTR-derived subgenomic flavivirus RNA (sfRNA) is absent from the WNV-poly(A) virus. Because the sfRNA is associated with modulation of the host interferon response, the authors hypothesize that the virus is sensitive to interferon. They show this experimentally in culture by addition of exogenous interferon, as well as in vivo,

showing greater lethality of the WNV-poly(A) virus in IFNAR^{-/-} mice.

Comments/ concerns:

1. In general, the data support the authors conclusions. However, they suggest several times that this strategy can be applied to all mosquito-borne flaviviruses. This is certainly a reasonable hypothesis, but it has not been tested, since data is only shown for WNV. Notably, small (30 nt) deletions in the 3' UTR of multiple dengue virus serotypes decrease expression of the sfRNA for all these viruses, but the extent of attenuation differs for the different serotypes. The use of polyA substitution for the 3'UTR should therefore be the focus of a speculative paragraph in the discussion.

Response: Many thanks for your comments. Since we did not test our method in other flaviviruses, we changed the title of the manuscript by replacing “flaviviruses” with “WNV” and tuned down some statement in the text of manuscript. We also re-organized the discussion part of our manuscript, and focus on WNV for discussion. Additionally, we mentioned the possibility applied to other flaviviruses need to be tested in future study.

2. It would strengthen the manuscript is the authors transfected RNAs with defined poly(A) lengths (rather than a pool of multiple lengths) to define what the required length of insert is.

Response: Many thanks for your comments. In fact, in our original manuscript we have determined the poly(A) is around 130nt for pooled viruses and described in Line 166 as “around 130 nt of poly(A) tract between NS5 coding region and 3'- sHP-SL”. In our revised manuscript, we selected plaque purified viruses for animal study. The plaque purified virus also contains around 130nt poly(A) tracts similar to that of pooled passage viruses, and also attenuated in mice and protected mice from WT WNV challenge. The new data were presented in the revised manuscript as Fig.1C,D and Appendix Figure S7

3. What is the red arrow in Fig 2D? The text refers to black, yellow, and blue arrows, but not red. It seems to be an apoptotic cell, although examination of this figure requires significant zooming in on a computer screen.

Response: Thank you very much for your comments. In our original manuscript's Fig 2d, we missed the description of the red arrow. We are very sorry for such mistake. The red arrow is used to indicate necrotic neurons and we added its description in Fig 2D in our revised manuscript. Moreover, we re-organized Fig 2 with a magnified view of Fig 2D.

4. The T cell data in figures 3g and 3h are confusing. The PMA+ionomycin treatment should induce all T-cells capable of secreting IFN γ to do so. The results in 3h suggest that the majority of T-cells in the spleens of vaccinated mice are responding to WNV peptides- this seems high? This experiment should also include an irrelevant (non-WNV) peptide control.

Response: Many thanks for your comments. In our original manuscript (3g and 3h), we mixed mock and poly(A) between PMA+IONO treatment and NS4B+E treatment. We are very sorry for such careless mistake. We performed new ELISPOT experiment including an irrelevant scramble peptide as a control. In our revised manuscript, new data were presented as Fig. 3G and Fig. 3H to replace old data.

Minor points:

5. The manuscript should be looked over for standard English grammar and usage (mostly missing indefinite/ definite articles). Line 168: "did be" should be changed to "was" or "is."

Response: Corrected.

25th Jun 2021

Dear Prof. Zhang,

Thank you for the submission of your manuscript to EMBO Molecular Medicine. I am pleased to inform you that we will be able to accept your manuscript pending the following final amendments:

- 1) Figures: In Figure 2D please add origin boxes of magnifications for mock and poly(A) samples.
- 2) In the main manuscript file, please do the following:
 - Correct/answer the track changes suggested by our data editors by working from the attached document.
 - Remove text highlight colour.
 - Make sure that all special characters display well.
 - Add callout for Figure 4D.
 - Rename "Declaration of interest" to "Conflict of interest".
 - In M&M, provide the antibody dilutions that were used for each antibody.
 - In M&M, provide primer sequences.
 - In M&M, a statistical paragraph should reflect all information that you have filled in the Authors Checklist, especially regarding randomization, blinding, replication.
 - Provide data availability statement. If no data are deposited in public repositories, please add the sentence: "This study includes no data deposited in external repositories".

Please check "Author Guidelines" for more information.

<https://www.embopress.org/page/journal/17574684/authorguide#availabilityofpublishedmaterial>

- 3) Funding: Please include all sources of funding in the manuscript and our submission system.
- 4) Synopsis:
 - Synopsis text: Please revise your synopsis text for grammar and syntax. Provide a short stand first (maximum of 300 characters, including space) as well as 2-5 one sentence bullet points that summarise the paper. Please write the bullet points to summarise the key NEW findings. They should be designed to be complementary to the abstract - i.e. not repeat the same text. We encourage inclusion of key acronyms and quantitative information (maximum of 30 words / bullet point). Please use the passive voice.
 - Synopsis image: Please check your synopsis image, revise it if necessary and submit the final versions with your revised manuscript. Please be aware that in the proof stage minor corrections only are allowed (e.g., typos).
- 5) For more information: There is space at the end of each article to list relevant web links for further consultation by our readers. Could you identify some relevant ones and provide such information as well? Some examples are patient associations, relevant databases, OMIM/proteins/genes links, author's websites, etc...
- 6) As part of the EMBO Publications transparent editorial process initiative (see our Editorial at <http://embomolmed.embopress.org/content/2/9/329>), EMBO Molecular Medicine will publish online a Review Process File (RPF) to accompany accepted manuscripts. This file will be published in conjunction with your paper and will include the anonymous referee reports, your point-by-point response and all pertinent correspondence relating to the manuscript. Let us know whether you agree with the publication of the RPF and as here, if you want to remove or not any figures from it prior to publication. Please note that the Authors checklist will be published at the end of the RPF.
- 7) Please provide a point-by-point letter INCLUDING my comments as well as the reviewer's reports and your detailed responses (as Word file).

I look forward to reading a new revised version of your manuscript as soon as possible.

Yours sincerely,

Zeljko Durdevic

***** Reviewer's comments *****

Referee #1 (Remarks for Author):

Authors have satisfactorily answer my comments and, in my opinion, most of those of the other reviewers. They have conducted different experiments to confirm their results, and when no, authors have gave convincing arguments to not do it.

Referee #3 (Remarks for Author):

The revised version of this manuscript addresses my previous concerns.

The authors performed the requested editorial changes.

25th Jun 2021

Dear Prof. Zhang,

Thank you for the submission of your manuscript to EMBO Molecular Medicine. I am pleased to inform you that we will be able to accept your manuscript pending the following final amendments:

1) Figures: In Figure 2D please add origin boxes of magnifications for mock and poly(A) samples.

Response: We have added origin boxes of magnifications for mock and poly(A) samples in Figure 2D.

2) In the main manuscript file, please do the following:

- Correct/answer the track changes suggested by our data editors by working from the attached document.
- Remove text highlight colour.
- Make sure that all special characters display well.
- Add callout for Figure 4D.
- Rename "Declaration of interest" to "Conflict of interest".
- In M&M, provide the antibody dilutions that were used for each antibody.
- In M&M, provide primer sequences.
- In M&M, a statistical paragraph should reflect all information that you have filled in the Authors Checklist, especially regarding randomization, blinding, replication.
- Provide data availability statement. If no data are deposited in public repositories, please add the sentence: "This study includes no data deposited in external repositories".

Please check "Author Guidelines" for more

information. <https://www.embopress.org/page/journal/17574684/authorguide#availabilityofpublishedmaterial>

Response: All are done as above listed.

3) Funding: Please include all sources of funding in the manuscript and our submission system.

Response: The sources of funding were added in Acknowledge.

4) Synopsis:

- Synopsis text: Please revise your synopsis text for grammar and syntax. Provide a short stand first (maximum of 300 characters, including space) as well as 2-5 one sentence bullet points that summarise the paper. Please write the bullet points to summarise the key NEW findings. They should be designed to be complementary to the abstract - i.e. not repeat the same text. We encourage inclusion of key acronyms and quantitative information (maximum of 30 words / bullet point). Please use the passive voice.

- Synopsis image: Please check your synopsis image, revise it if necessary and submit the final versions with your revised manuscript. Please be aware that in the proof stage minor corrections only are allowed (e.g., typos).

Response: We have revised our synopsis text as required.

5) For more information: There is space at the end of each article to list relevant web links for further consultation by our readers. Could you identify some relevant ones and provide such information as well? Some examples are patient associations, relevant databases, OMIM/proteins/genes links, author's websites, etc...

Response: Two web links about WNV introduction were added in our revised manuscript.

6) As part of the EMBO Publications transparent editorial process initiative (see our Editorial at <http://embomolmed.embopress.org/content/2/9/329>), EMBO Molecular Medicine will publish online a Review Process File (RPF) to accompany accepted manuscripts. This file will be published in conjunction with your paper and will include the anonymous referee reports, your point-by-point response and all pertinent correspondence relating to the manuscript. Let us know whether you agree with the publication of the RPF and as here, if you want to remove or not any figures from it prior to publication. Please note that the Authors checklist will be published at the end of the RPF.

Response: We agree with the publication of the RPF.

7) Please provide a point-by-point letter INCLUDING my comments as well as the reviewer's reports and your detailed responses (as Word file).

Response: Response was prepared as suggested.

I look forward to reading a new revised version of your manuscript as soon as possible.

Yours sincerely,

Zeljko Durdevic

***** Reviewer's comments *****

Referee #1 (Remarks for Author):

Authors have satisfactorily answer my comments and, in my opinion, most of those of the other reviewers. They have conducted different experiments to confirm their results, and when no, authors have gave convincing arguments to not do it.

Response: We are gratefull for your comments and review of our manuscript.

Referee #3 (Remarks for Author):

The revised version of this manuscript addresses my previous concerns.

Response: We are gratefull for your comments and review of our manuscript.

We are pleased to inform you that your manuscript is accepted for publication and is now being sent to our publisher to be included in the next available issue of EMBO Molecular Medicine.

Corresponding Author Name: Han-Qing Ye and Bo Zhang

Manuscript Number: EMM-2021-14108